# ZERO-ORDER DIFFUSION GUIDANCE FOR INVERSE PROBLEMS

## ABSTRACT

We propose zero order diffusion guidance, a method that allows using a diffusion model to solve inverse problems without access to the gradients of the process we seek to invert. Our method employs a zero-order gradient estimator combined with a novel differentiable dimensionality reduction strategy to approximate true gradients during guidance while keeping the task computationally tractable in thousands of dimensions. We apply our method to model inversion and demonstrate how it can be used to reconstruct high-quality faces in a realistic scenario where the adversary has only black-box access to face embeddings. Across a range of inverse problems—including synthetic experiments and JPEG restoration—we show that access to gradients is not necessary for effective guidance. Our black-box method matches white-box performance, thus expanding the scope of inverse problems that can be solved with diffusion-based approaches.

## 1 INTRODUCTION

Diffusion models have emerged as powerful generative models capable of generating realistic images, audio, and text (Rombach et al., 2021). Central to their success is their controllability, i.e., their ability to use gradient information from a differentiable guidance function to steer the output towards a set of desirable properties (Ho & Salimans, 2022). This capability is important for solving inverse problems (e.g., colorization (Saharia et al., 2022), MRI processing (Kazerouni et al., 2023), 3D reconstruction (Anciukevičius et al., 2023)) as well as for generative design (e.g., producing biological molecules with desired properties) (Chung & Ye, 2022; Rombach et al., 2021; 2022; Lugmayr et al., 2022; Poole et al., 2022).

However, obtaining gradient information useful for guidance may be challenging. In many inverse problems, the process that we seek to invert may be complex, non-differentiable, or available only as a black box. For example, in security applications, we may be working with sensitive information from data obfuscated by a black-box program (Duong et al., 2020). In generative design settings, the data we work with may be discrete (e.g., language, proteins, DNA), and thus the underlying guidance process does not have gradients (Sahoo et al., 2024; Austin et al., 2021).

Here, we introduce zero order diffusion guidance, an algorithm that only assumes input-output access to a guidance function, and not its gradients. Our key idea is to approximate classifier-based guidance (Dhariwal & Nichol, 2021) by using zero-order gradient estimators.

Out of the box, zero-order methods are too slow for guidance: we propose two types of estimators combined with novel dimensionality reduction and error correction strategies that make zero-order methods practical for classifier-based guidance without an excessive increase in computation time. The resulting method **Z**ero **OR**der **O**ptimization for classifier guidance (ZORO) is simple, easy-to-implement, and matches the performance of white-box guidance.

We demonstrate the applicability of ZORO across two application areas: black-box inverse problems and non-differentiable guidance. We first use our method to study the security of face recognition systems and propose a strategy for inverting a black-box face embedding model to recover human faces from their embeddings (Duong et al., 2020). We show that zero order guidance with a custom sampler yields an inversion attack that is both effective (i.e., it recovers high-quality images) and easier to stage than previous, GAN-based, black-box inversion methods, and that sheds light on vulnerabilities of face embedding models. Additionally, we utilize ZORO for JPEG restoration (Li

Figure 1: Non-cherry picked images recovered by ZORO from the test set of FFHQ dataset. We are able to invert the outputs of ElasticFace face embedding models and produce reconstructions with high embedding-space similarity to the ground-truth images. The numbers at the top denote the cosine similarity between the embeddings of the reconstructed image and the ground truth image.

& Wand, 2016; Yang et al., 2020; Yu et al., 2022; Si & Kim, 2024)—a task where the underlying gradient is undefined—and demonstrate that it outperforms all baseline methods.

**Contributions** In summary, our work makes the following contributions:

1. We introduce ZORO, a method for controlling the output of diffusion models in settings where we only have black-box access to the guidance process.

2. We propose an effective way to apply zero-order gradient estimators to higher dimensional data that relies on novel dimensionality reduction and error correction strategies.

3. We demonstrate across a range of inverse problems that exact gradient information is not required for effective guidance, and that our black-box method matches white-box performance, including for the task of inverting face embeddings or restoring JPEG images.

## 2 BACKGROUND

### 2.1 DIFFUSION MODELS

Given a sample $\mathbf{x}_0 \in \mathbb{R}^n$ from the data distribution $\mathcal{D}$, a forward diffusion process is defined as a Markov chain of latent variables $\mathbf{x}_1, \ldots, \mathbf{x}_T$ that progressively add noise to $\mathbf{x}_0$. A reverse diffusion process starts from samples of $\mathbf{x}_T \sim p_T$ and seeks to recover samples $\mathbf{x}_0 \sim p_0$. Examples of forward processes include Gaussian diffusion, masking diffusion, and uniform noise.

**Gaussian Diffusion** For example, we may add Gaussian noise of standard deviation $\sigma_t \in \mathbb{R}^+$ (with $\sigma_t$ monotonically increasing in $t$), i.e. $p_t(\mathbf{x}|\mathbf{x}_0) = \mathcal{N}(\mathbf{x}_0, \sigma_t^2 \mathbf{I}_n)$ where $p_t(.)$ denotes the distribution of the sample $\mathbf{x}_t$ at timestep $t$. The forward diffusion process can be modeled by the SDE $d\mathbf{x} = \sqrt{\frac{d[\sigma_t^2]}{dt}} d\mathbf{w}$, where $d\mathbf{w}$ is the standard Wiener process (a.k.a Brownian motion) and $dt$ is an infinitesimal negative timestep. Anderson (1982) states that the reverse of a diffusion process is given by the reverse-time SDE:

$$d\mathbf{x} = \left[ -\frac{d[\sigma_t^2]}{dt} \nabla_{\mathbf{x}} \log p_t(\mathbf{x}) \right] dt + \sqrt{\frac{d[\sigma_t^2]}{dt}} d\overline{\mathbf{w}} \tag{1}$$

Once the score of each marginal distribution, $\nabla_{\mathbf{x}} \log p_t(\mathbf{x})$, is known for all $t$, we can derive the reverse diffusion process from (1) and simulate it to sample from $p_0$. Since $\nabla_{\mathbf{x}} \log p_t(\mathbf{x})$ is unknown during the reverse diffusion process, we parameterize it using a neural network $\mathbf{s}_\theta(\mathbf{x}; \sigma_t) : \mathbb{R}^n \to \mathbb{R}^n$ trained with the score matching objective (Karras et al., 2022):

$$\mathcal{L} = \mathbb{E}_{\mathbf{x}_0 \sim \mathcal{D}, t \sim \{0,\ldots,T\}, \tilde{\mathbf{x}} \sim p_t(.|\mathbf{x}_0)} \|\mathbf{s}_\theta(\tilde{\mathbf{x}}; \sigma_t) - \nabla_{\tilde{\mathbf{x}}} \log p_t(\tilde{\mathbf{x}}|\mathbf{x}_0)\|_2^2. \tag{2}$$

Then, samples are drawn from diffusion models by solving the SDE in (1), such as with Euler's method, Euler-Maruyama, and higher order SDE solvers (Lu et al., 2022; Karras et al., 2022).

## 2.2 Diffusion Model Guidance

**Classifier-based guidance**   To draw samples from the conditional distribution $p_t(\mathbf{x}|\mathbf{c})$, Dhariwal & Nichol (2021) seek to train a separate classifier $p(\mathbf{c}|\mathbf{x})$ alongside the diffusion model $\mathbf{s}_\theta$, where $\mathbf{c}$ is defined to be conditional information that guides the generation process. Then the samples from $p(\mathbf{x}|\mathbf{c})$ can be drawn by using the following score function in the reverse process:

$$
\begin{aligned}
\nabla_{\mathbf{x}} \log p_t(\mathbf{x}|\mathbf{c}) &= \nabla_{\mathbf{x}} \log p_t(\mathbf{x}) + \nabla_{\mathbf{x}} \log p_t(\mathbf{c}|\mathbf{x}) \\
&\approx \mathbf{s}_\theta(\mathbf{x}, t) + \alpha_{\mathrm{c}} \nabla_{\mathbf{x}} \log q_\phi(\mathbf{c}|\mathbf{x}),
\end{aligned}
\tag{3}
$$

where $q_\phi(\mathbf{c}|\mathbf{x})$ is the probability of attribute $\mathbf{c}$ given input $\mathbf{x}$, parameterized by a network with parameters $\phi$ and the scalar $\alpha_{\mathrm{c}} \in \mathbb{R}^+$ scales up the strength of the classifier gradients.

**Classifier-free guidance**   Ho & Salimans (2022) show that samples can be drawn from $p_t(\mathbf{x}|\mathbf{c})$ without training a separate classifier. For this, they condition the score model $\mathbf{s}_\theta$ on $\mathbf{c}$ during training. Furthermore, they replace $\mathbf{c}$ with $\emptyset$ randomly with a probability of $10\%$ during training. During sampling they use the following approximation of the score function:

$$
\begin{aligned}
\nabla_{\mathbf{x}} \log \tilde{p}_t(\mathbf{x}|\mathbf{c}) &\approx (1 - \alpha_{\mathrm{f}}) \nabla_{\mathbf{x}} \log p_\theta(\mathbf{x}) + \alpha_{\mathrm{f}} \nabla_{\mathbf{x}} \log p_\theta(\mathbf{x}|\mathbf{c}) \\
&= (1 - \alpha_{\mathrm{f}}) \mathbf{s}_\theta(\mathbf{x}, \emptyset, t) + \alpha_{\mathrm{f}} \mathbf{s}_\theta(\mathbf{x}, \mathbf{c}, t).
\end{aligned}
\tag{4}
$$

Where $\alpha_{\mathrm{f}}$ is a hyperparameter that controls the amount of guidance added, $\nabla_{\mathbf{x}} \log p_\theta(\mathbf{x})$ is the gradient of the log of the unconditional probability , and $\nabla_{\mathbf{x}} \log p_\theta(\mathbf{x}|\mathbf{c})$ is the gradient of the log of the conditional probability.

**Inverse Problems**   Inverse problems seek to recover $\mathbf{x}_0 \in \mathbb{R}^n$ from a measurement $\mathbf{c} \in \mathbb{R}^m$ generated by function $f : \mathbb{R}^n \to \mathbb{R}^m$ such that $\mathbf{c} = f(\mathbf{x}_0)$. Let $\ell(\mathbf{c}, f(\mathbf{x}_0))$ be the loss between $f(\mathbf{x}_0)$ and $\mathbf{c}$ (e.g., $\|\mathbf{c} - f(\mathbf{x}_0)\|_2^2$); this implicitly defines an energy-based distribution $q_\phi(\mathbf{c}|\mathbf{x}_0) = \exp\left(-\ell(\mathbf{c}, f(\mathbf{x}_0))\right)/\mathcal{Z}$, with normalizing constant $\mathcal{Z}$. Inverse problems can be solved using classifier-based guidance (3) with the following conditional score:

$$
\nabla_{\mathbf{x}} \log q_\phi(\mathbf{c}|\mathbf{x}) = -\mathbb{E}_{\tilde{\mathbf{x}}_0 \sim p_\theta(.|\mathbf{x})} \left[ \nabla_{\tilde{\mathbf{x}}_0} \ell(\mathbf{c}, f(\tilde{\mathbf{x}}_0)) \right],
\tag{5}
$$

where $\tilde{\mathbf{x}}_0$ is computed using the Tweedie's formula $\tilde{\mathbf{x}}_0 = \mathbf{x} + \sigma_t^2(t) \mathbf{s}_\theta(\mathbf{x}, t)$ (Efron, 2011). Note that if $f$ is not bijective, we simply seek the closest $\mathbf{x}_0$ that produced $\mathbf{c}$ as defined by our loss.

## 3 Zero-Order Diffusion Guidance

In this paper, we are interested in extending diffusion guidance to functions $f$ for which a gradient is not available. In security applications, we may be working with sensitive information from data obfuscated by a black-box program. In other settings, the data may be discrete (e.g., language, proteins, DNA), and thus the underlying guidance process does not have gradients.

Our key idea is to approximate classifier-based guidance by using zero-order gradient estimators. Specifically, we approximate the guidance term $\nabla_{\mathbf{x}} \log q_\phi(\mathbf{c}|\mathbf{x})$ in (3) with a zero-order approximation $\hat{\nabla}_{\mathbf{x}} \log q_\phi(\mathbf{c}|\mathbf{x})$ in Sec. 3.1. Out of the box, zero-order methods are slow: to make them practical, we develop novel dimensionality reduction and error correction strategies in Sec. 3.2.

### 3.1 Zero-Order Gradient Estimation

Given a guidance term expressed in the form of a loss function $\ell(\mathbf{x}) : \mathbb{R}^n \to \mathbb{R}^+$ of an input $\mathbf{x}$, we aim to perform guidance without relying on the gradients of $\ell$. Our strategy will be to define an approximate gradient operator $\hat{\nabla}_{\mathbf{x}}$ that estimates the true gradient $\nabla_{\mathbf{x}}$.

One approach to approximate $\hat{\nabla}_{\mathbf{x}} \ell(\mathbf{x})$ is be to use a scaled random gradient estimator: $\hat{\nabla}_{\mathbf{x}} \ell(\mathbf{x}) = \sum_{i=1}^k \frac{\ell(\mathbf{x} + \beta \mathbf{u}) - \ell(\mathbf{x})}{\beta} \mathbf{u}$, where $k$ is the number of samples, $\beta > 0$ is a smoothing parameter, and $\mathbf{u}$ is a vector drawn from a unit Euclidean sphere. As $\beta \to 0^+$ and $k \to \infty$, this quantity tends to the true gradient, i.e., $\hat{\nabla}_{\mathbf{x}} \ell(\mathbf{x}) = \nabla_{\mathbf{x}} \ell(\mathbf{x})$, as $\beta \to 0^+$ and $k \to \infty$. However, this method is known to suffer from high variance and slow convergence (Liu et al., 2020).

In this work we propose using a coordinate-wise gradient estimation technique. Specifically, we approximate the partial derivative of the $i$-th component of $\mathbf{x}$, $\frac{\partial f(\mathbf{x})}{\partial \mathbf{x}_i}$, individually:

$$\hat{\nabla}_{\mathbf{x}}\ell(\mathbf{x}) = \sum_{i=1}^{n} \frac{\ell(\mathbf{x} + \delta \mathbf{e}_i) - \ell(\mathbf{x})}{\delta} \mathbf{e}_i, \tag{6}$$

where $\delta > 0$ is a small constant, and $\mathbf{e}_i$ is a standard basis vector with only the $i$-th component equal to 1. It is clear that $\hat{\nabla}_{\mathbf{x}}\ell(\mathbf{x})$ converges to $\nabla_{\mathbf{x}}\ell(\mathbf{x})$ as $h \to \infty$ when $\ell$ is differentiable. Therefore, this approach approximates the true gradient with $n$ queries, while the direction-wise approach requires infinitely many samples to achieve the same accuracy. When $\mathbf{x}$ is discrete but admits a continuous relaxation over $\mathbb{R}^n$, this method also approximates the gradient in the extended space. In Suppl. C.2, we provide a detailed analysis of both methods and demonstrate the effectiveness of the proposed coordinate-wise method over the direction-wise gradient approximation technique.

### 3.2 SCALING UP ZERO-ORDER GRADIENT ESTIMATION

Note that computing $\hat{\nabla}_{\mathbf{x}}\ell(\mathbf{x})$ using the estimate in (6) requires $O(n)$ queries, one for each coordinate. This is often impractical; for instance, if the input to $\ell$ is a $224 \times 224 \times 3$ image, approximating $\hat{\nabla}_{\mathbf{x}}\ell(\mathbf{x})$ will require $\approx 150K$ queries. Moreover, recall that sampling from a diffusion model involves $T$ denoising steps. This presents a significant challenge for zero-order guidance, where gradient guidance is applied at each timestep in the reverse diffusion process, meaning that we would have to make $\approx 150,000T$ queries during sampling.

Surprisingly, we find that a simple differentiable downsampling function can resolve the above challenge when combined with a novel error correction strategy which we present in Sec. 3.2.2. Prior works (Tu et al., 2020) have used a deep autoencoder to reduce dimensionality and perform zero-order guidance in the latent space. In contrast, our approach doesn't require training an additional neural network and in Suppl. C.3 we demonstrate that our proposed approach significantly outperforms a deep autoencoder.

### 3.2.1 DOWNSAMPLING

Specifically, we use a pair of downscaling and upscaling functions, $g : \mathbb{R}^n \to \mathbb{R}^m$ and $h : \mathbb{R}^m \to \mathbb{R}^n$, where $g$ is "bilinear downsampling" and $h$ is "bilinear upsampling" such that $x \approx h(g(\mathbf{x}))$, where $m < n$. The function $g$ reduces the data from $n$ to $m$ dimensions, making zero-order guidance in the latent space more computationally efficient.

Our proposed downscaling method works in the following manner. We first create a low-resolution downsampled image, $\mathbf{y} = g(\mathbf{x})$. We then approximate the loss gradient, $\ell(h(\mathbf{y}))$, with respect to the smaller downsampled image by computing $\hat{\nabla}_{\mathbf{y}}\ell(h(\mathbf{y}))$ using our zero-order method. The new gradient approximation formula is given by,

$$\hat{\nabla}_{\mathbf{y}}\ell(h(\mathbf{y})) = \sum_{i=1}^{m} \frac{\ell(h(\mathbf{y} + \delta \mathbf{e}_i)) - \ell(h(\mathbf{y}))}{\delta} \mathbf{e}_i. \tag{7}$$

Since, $g$ denotes a differentiable mapping, one can compute $\hat{\nabla}_{\mathbf{x}}\ell(\mathbf{x})$ by chain rule:

$$\hat{\nabla}_{\mathbf{x}}\ell(\mathbf{x}) = \left[ \frac{\partial \mathbf{y}}{\partial \mathbf{x}} \right]^{\top} \hat{\nabla}_{\mathbf{y}}\ell(h(\mathbf{y})). \tag{8}$$

### 3.2.2 ERROR CORRECTION

Note that the terms $\ell(h(\mathbf{y}))$ and $\ell(h(\mathbf{y} + \delta \mathbf{e}_i))$ play critical roles in the gradient estimate (7). However, both $h(\mathbf{y})$ and $h(\mathbf{y} + \delta \mathbf{e}_i)$ are susceptible to artifacts introduced by the lossy upsampling operation $h$ and the perturbation $\delta \mathbf{e}_i$. We found this issue to be particularly problematic in our facial embedding inversion experiments, where the approximate gradient $\hat{\nabla}_{\mathbf{y}}\ell(\mathbf{y})$ differed significantly from the true gradient $\nabla_{\mathbf{y}}\ell(h(\mathbf{y}))$. This discrepancy arises because the loss function—which involves the embedding model—is especially sensitive to such artifacts.

While the artifacts introduced by the perturbation $\delta \mathbf{e}_i$ are unavoidable, we propose a method to mitigate the artifacts caused by the lossy reconstructions from $h$. Specifically, we introduce an error correction term, defined as $\Delta = \mathbf{x} - h(\mathbf{y})$. If the upsampling function $h$ can recover $\mathbf{x}$ from $\mathbf{y}$, then $\Delta = 0$. By adding this correction term, we reduce the impact of artifacts introduced by $h$ and improve the accuracy of our gradient estimates. The modified gradient approximation is then:

$$\hat{\nabla}_{\mathbf{y}} \ell(h(\mathbf{y})) = \sum_{i=1}^{m} \frac{\ell(h(\mathbf{y} + \delta \mathbf{e}_i) + \Delta) - \ell(h(\mathbf{y}) + \Delta)}{\delta} \mathbf{e}_i$$

$$= \sum_{i=1}^{m} \frac{\ell(h(\mathbf{y} + \delta \mathbf{e}_i) + \Delta) - \ell(\mathbf{x})}{\delta} \mathbf{e}_i. \tag{9}$$

### 3.3 ZORO: ZERO-ORDER OPTIMIZATION FOR CLASSIFIER-BASED GUIDANCE

**Motivating Example.** Consider the specific case where $\mathbf{c}$ is the output of a face embedding model $f$, and $\mathbf{x}$ is an unknown image of a face. In this setting, searching for the image with an embedding that minimizes the loss $\ell(f(\mathbf{x}), \mathbf{c}) = -\cos(f(\mathbf{x}), \mathbf{c})$ should, in theory, let us recover the face from its embedding. Classifier-based diffusion guidance has been quite successful at solving such inverse problems (Rombach et al., 2021; Chung et al., 2022a). However, this specific task of facial embedding inversion is more challenging because (3) isn't directly applicable. This is due to the intractability of computing the classifier gradients $\nabla_{\mathbf{x}} \log q_\phi(\mathbf{c}|\mathbf{x}) = \nabla_{\mathbf{x}} \cos(f(\mathbf{x}), \mathbf{c})$ when we only have black-box access to the embedding model $f(\mathbf{x})$.

**Zero-Order Classifier Guidance.** To address the aforementioned challenge, we propose ZORO, that uses Zero ORder Optimization for classifier guidance in diffusion models. Our method approximates the classifier gradients using zero-order methods as detailed in Equation 9. ZORO combines zero order classifier-based guidance with classifier-free guidance as follows:

$$\nabla_{\mathbf{x}} \log p_t(\mathbf{x}|\mathbf{c}) \approx \alpha_u \nabla_{\mathbf{x}} \log p_\theta(\mathbf{x}) + \alpha_f \nabla_{\mathbf{x}} \log p_\theta(\mathbf{x}|\mathbf{c}) + \alpha_c \hat{\nabla}_{\mathbf{x}} \log q_\phi(\mathbf{c}|\mathbf{x}), \tag{10}$$

where $\alpha_u, \alpha_f, \alpha_c \in \mathbb{R}$ denote the strengths of the unconditional, classifier-free, and classifier-based guidance terms respectively and $\hat{\nabla}_{\mathbf{x}} \log q_\phi(\mathbf{c}|\mathbf{x})$ denotes the zero order approximation of the true gradient $\nabla_{\mathbf{x}} \log q_\phi(\mathbf{c}|\mathbf{x})$. This formulation allows for the simultaneous use of both traditional classifier-free guidance and our zero order classifier-based guidance, without the need to compute any gradients. Our approach deviates from the conventional diffusion guidance methods, which typically use either classifier-based or classifier-free guidance exclusively. As shown in Table 6, we empirically demonstrate that combining both types of guidance is essential for achieving optimal performance.

## 4 EXPERIMENTAL SETUP

Our experiments can be divided into three parts: (1) demonstrating our method's effectiveness in a suite of synthetic tasks (4.1) and proving its real-world effectiveness in recovering faces from embeddings (4.2) as well as removing JPEG compression artifacts (4.3).

**Main metrics.** We measure ground-truth image generations using several classes of metrics. Our main metric is the cosine similarity $\cos(\mathbf{c}, f(\mathbf{x}'))$ between the ground-truth embedding $\mathbf{c}$ and the embedding $f(\mathbf{x}')$ of the generated image $\mathbf{x}'$. We also measure PSNR (Peak Signal-to-Noise Ratio), SSIM (Structural Similarity Index Measure), and LPIPS (Learned Perceptual Image Patch Similarity). As a measure of algorithm cost, we track *query count*, the total number of forward passes made with black-box model $f$. In gradient estimation experiments we measure the cosine similarity between the white-box (true) gradient and our black-box estimate.

**Baselines.** We compare the performance of ZORO to an *oracle*, where we have white-box access to the gradient $\nabla_{\mathbf{x}} \ell(\mathbf{x})$ and use (10) to draw samples from the diffusion model. This represents the upper bound of our method's performance. ZORO and the *oracle* share the same configurations for $\alpha_f$ and $\alpha_c$.

Table 1: Main results for synthetic experiments. We assess our method using PSNR, SSIM, MSE, and LPIPS, demonstrating that it consistently matches *oracle* performance.

| Corruption Type | Method | PSNR ($\uparrow$) | SSIM ($\uparrow$) | LPIPS ($\downarrow$) |
|---|---|---|---|---|
| Gaussian Blur & Grayscale | *oracle* | $18.10_{\pm 0.11}$ | $0.62_{\pm 0.00}$ | $0.12_{\pm 0.00}$ |
| | ZORO (Ours) | $17.84_{\pm 0.11}$ | $0.59_{\pm 0.00}$ | $0.13_{\pm 0.00}$ |
| Gaussian Blur & Masking | *oracle* | $16.74_{\pm 0.11}$ | $0.59_{\pm 0.00}$ | $0.08_{\pm 0.00}$ |
| | ZORO (Ours) | $16.61_{\pm 0.10}$ | $0.57_{\pm 0.00}$ | $0.08_{\pm 0.00}$ |
| Grayscale & Masking | *oracle* | $15.60_{\pm 0.12}$ | $0.73_{\pm 0.00}$ | $0.12_{\pm 0.00}$ |
| | ZORO (Ours) | $15.42_{\pm 0.10}$ | $0.66_{\pm 0.00}$ | $0.13_{\pm 0.00}$ |

**Diffusion Model.** The model architecture is based on the UNet of Song et al. (2020). We train a diffusion model to approximate $\mathbf{x}_0$ in the reverse process using a neural network, $\mathrm{D}_\theta(\mathbf{x}_t, t)$ : $[\mathbb{R}^n \times \mathbb{R}^+] \to \mathbb{R}^n$, with parameters $\theta$ and the corresponding the score function for the Variance Exploding type diffusion model in consideration is given by $\mathbf{s}_\theta(\mathbf{x}_t, t) = -(\mathbf{x}_t - \mathrm{D}_\theta(\mathbf{x}_t, t))/\sigma_t^2$ (Song et al., 2020) and trained using (2). For more details refer Suppl. B. Given our score function in (10), we solve (1) using the Heun sampler as proposed in Karras et al. (2022). The exact algorithm of the sampler is provided in Suppl. A. We set $T = 100$ for diffusion timesteps during sampling for all our experiments. More details can be found in Suppl. B.

**Training details.** We generally follow the training recipe from Karras et al. (2022) for training a diffusion model on the FFHQ dataset (Karras et al., 2019), which consists of $70,000$ images of human faces released under the Creative Commons license; as is standard in the literature, we split our data into $60,000$ train and $10,000$ test images. Note that we do not need paired facial data, only one face per individual, as we are not training a facial embedding model. We downsample images from their original resolution of $1024 \times 1024$ to $64 \times 64$ for training. We use a batch size of 256, a learning rate of $2e - 4$ with linear warm-up. We use the AdamW optimizer (Loshchilov & Hutter, 2019) and train for $200M$ images on 8 NVIDIA V100 GPUs, which takes about six days in total.

**ZORO.** Unless otherwise stated, ZORO uses Coordinate-Wise Zero-Order estimation with $\delta = 3.0$ (Sec. 3.1) and bilinear-downsampling to $48 \times 48$ resolution for downscaling (Sec. 3.2) as default configurations. We justify the choice of these hyperparameters in Sec. 4.4. Following Ho & Salimans (2022), we always set $\alpha_u = 1 - \alpha_f$ and $\alpha_f, \alpha_c$ are task-specific.

## 4.1 SYNTHETIC EXPERIMENTS

We begin by evaluating the performance of ZORO on a suite of synthetic tasks. We observe that **ZORO nearly matches the performance of *oracle*** qualitatively and quantitatively as shown in Sec. 1. Specifically, we focus on simple corruption functions, where each function is a composition of standard corruptions such as color removal, blurring, and pixel masking. The goal is to recover the clean input underlying the corrupt input with just a black box access to the corruption function.

For informative experiments, we design the corruption function $f(\mathbf{x})$ to be non-differentiable, and lacking a trivial inverse $f^{-1}(\mathbf{x})$. Based on these criteria, we construct the following compositional corruptions: (1) Gaussian Blur & Grayscale, (2) Gaussian Blur & Mask, and (3) Grayscale & Mask. We implement these corruption functions using Kornia (Riba et al., 2020), which provides a broad set of differentiable transformations essential for constructing the *oracle*.

It's important to note that the diffusion model was not specifically trained to reverse these corruptions, so we perform guidance with $\alpha_f = 0$. Upon examining the reconstructions, we find that optimal guidance settings are $\alpha_f = 500$ for corruption types (1) and (2), and $\alpha_f = 1000$ for (3).We provide visualizations of the ground truth image, the corrupted image, and their reconstructions for corruption types (1), (2), and (3) in Fig. 8, Fig. 9, and Fig. 7, respectively.

Table 2: Main results for real-world experiment inverting face embeddings on the test set of FFHQ dataset. We assess our method using PSNR, SSIM, and LPIPS, demonstrating that it consistently matches oracle performance. Error bars indicate the standard deviation.

| Method | **Emb** (↑) | PSNR (↑) | SSIM (↑) | LPIPS (↓) |
|---|---|---|---|---|
| *oracle* | $0.84_{\pm 0.04}$ | $11.28_{\pm 1.76}$ | $0.36_{\pm 0.06}$ | $0.151_{\pm 0.00}$ |
| EDM (Karras et al., 2022) | $0.52_{\pm 0.12}$ | $8.99_{\pm 1.21}$ | $0.14_{\pm 0.03}$ | $0.226_{\pm 0.00}$ |
| Vendrow & Vendrow (2021) | $0.23$ | — | — | — |
| ID3PM (Kansy et al., 2023) | $0.65_{\pm 0.03}$ | $11.28_{\pm 1.75}$ | $0.35_{\pm 0.05}$ | $0.151_{\pm 0.00}$ |
| ZORO (Ours) | $\mathbf{0.79}_{\pm 0.04}$ | $\mathbf{11.28}_{\pm 1.75}$ | $\mathbf{0.36}_{\pm 0.06}$ | $\mathbf{0.151}_{\pm 0.00}$ |

## 4.2 FACIAL EMBEDDING INVERSION

We next evaluate our method in a real-world scenario, focusing on reconstructing facial images from their embeddings using a face embedding model. For this experiment, we selected the `ElasticFace-Cos` model, referred to as ElasticFace, from Boutros et al. (2022). ElasticFace is one of the most widely used open-source models for facial recognition and provides both an open-source implementation and pre-trained model weights. Like other popular models, ElasticFace is trained to maximize the cosine similarity between embeddings of facial images from the same individual.

To create our test dataset, we embedded all images from the FFHQ dataset using ElasticFace, resulting in pairs of original images and their corresponding embeddings $(\mathbf{x}, f(\mathbf{x}))$. We train our diffusion model to generate $\mathbf{x}$ conditioned on $f(\mathbf{x})$. Following Ho & Salimans (2022), we replace $f(\mathbf{x})$ with $\emptyset$ with 10% probability during training to aid classifier free guidance. In preliminary experiments on a small batch of images, we found that setting $\alpha_f = 10$ and $\alpha_c = 1000$ in (10) produced the best cosine similarity.

We compare the performance of our method against several baselines: the *oracle*, the classifier-based guidance diffusion approach of Kansy et al. (2023), the unconditional generation from a diffusion model as in Karras et al. (2022), and the embedding inversion method of Vendrow & Vendrow (2021), which uses a Generative Adversarial Network (Goodfellow et al., 2014). Detailed metrics are provided in Table 2. We observe that our method outperforms all the baselines and achieves results comparable to those of the *oracle*, even in this challenging setting.

In Table 6, we ablate different components of ZORO. First, we observe that using downsampling + $\delta$ correction (9) as a dimensionality reduction technique outperforms using an autoencoder. Next we separately analyze the effects of the classifier-free component, regulated by $\alpha_f$, and the classifier-based component, regulated by $\alpha_c$, in ZORO. Note that setting $\alpha_f = 0$ and $\alpha_c = 0$ corresponds to generation without guidance, which yields an embedding similarity of 0.52—representing the worst-case performance. With only classifier-free guidance (CFG), i.e., $\alpha_f = 10$ and $\alpha_c = 0$, the cosine similarity increases to 0.65. Similarly, with only classifier-based guidance (CBG), i.e., $\alpha_f = 0$ and $\alpha_c = 1000$, we achieve an embedding similarity of 0.72. However, when both CBG and CFG are combined, we attain the highest embedding similarity score of **0.79**.

## 4.3 JPEG RESTORATION

We consider the ability of ZORO to reverse the JPEG compression algorithm, which is differentiable. We consider the three JPEG quality factors (QFs) used in Si & Kim (2024)—5, 10, and 20—to restore degraded images from the FFHQ test set. In Table 3, we report quantitative results on JPEG restoration, comparing our method against several supervised learning algorithms (Li & Wand, 2016; Yang et al., 2020; Yu et al., 2022; wai, 2023), all of which are specifically trained for restoring JPEG images. Our goal in this experiment is to use a pretrained diffusion model trained on uncompressed FFHQ images and perform guidance using ZORO. In this context, $f$ represents the JPEG compression function, and we perform guidance by minimizing the mean squared error $\ell(f(\mathbf{x}'), c) = \|f(\mathbf{x}') - c\|_2^2$ between the degraded image $c$ and the predicted image $\mathbf{x}'$.

Table 3: Main results for JPEG corruption experiments on the test set of FFHQ. We assess our method using PSNR, SSIM, and LPIPS, demonstrating that it consistently improves the image quality. [†]Reported in Si & Kim (2024). Error bars indicate standard deviation.

|  | PSNR ($\uparrow$) | SSIM ($\uparrow$) | LPIPS ($\downarrow$) |
|---|---|---|---|
| ESRGAN[†] (Li & Wand, 2016) | $19.76_{\pm 0.71}$ | $0.54_{\pm 0.00}$ | $0.67_{\pm 0.00}$ |
| HiFaceGAN[†] (Yang et al., 2020) | $20.59_{\pm 0.70}$ | $0.60_{\pm 0.00}$ | $0.67_{\pm 0.00}$ |
| ESDNet-L[†] (Yu et al., 2022) | $17.26_{\pm 4.31}$ | $0.57_{\pm 0.00}$ | $0.62_{\pm 0.00}$ |
| waifu2x[†] (wai, 2023) | $20.58_{\pm 0.47}$ | $0.60_{\pm 0.00}$ | $0.64_{\pm 0.00}$ |
| Si & Kim (2024) | $21.68_{\pm 1.44}$ | $0.68_{\pm 0.00}$ | $0.37_{\pm 0.00}$ |
| ZORO (Ours) | $\mathbf{25.22_{\pm 1.42}}$ | $\mathbf{0.82_{\pm 0.00}}$ | $\mathbf{0.04_{\pm 0.00}}$ |

Table 4: Ablations for JPEG restoration across different quality factors.

| Method | Quality Factors | | | | | |
|---|---|---|---|---|---|---|
|  | 5 | | 10 | | 20 | |
|  | PSNR($\uparrow$) | SSIM ($\uparrow$) | PSNR($\uparrow$) | SSIM ($\uparrow$) | PSNR($\uparrow$) | SSIM ($\uparrow$) |
| Si & Kim (2024) | **21.61** | **0.67** | 21.68 | 0.67 | 21.68 | 0.68 |
| ZORO(ours) | $20.46_{\pm 1.03}$ | $0.64_{\pm 0.04}$ | $\mathbf{23.53_{\pm 1.03}}$ | $\mathbf{0.77_{\pm 0.04}}$ | $\mathbf{25.22_{\pm 1.3}}$ | $\mathbf{0.82_{\pm 0.03}}$ |

Notably, even though the model used in ZORO has never seen any JPEG images compressed to these quality factors, it outperforms all the baselines in Table 3, demonstrating the strength of zero-order guidance in task-agnostic diffusion models. In Table 4, we show that ZORO outperforms Si & Kim (2024) on QFs 10 and 20, while performing slightly worse on QF 5. Furthermore, we provide qualitative results in Fig. 2 (cherry-picked examples) and Suppl. 4 (randomly selected examples), where ZORO convincingly recovers the ground truth images from the degraded inputs.

## 4.4 GRADIENT ESTIMATION

In this section, we conduct comprehensive analyses to assess the behavior and performance of ZORO. Specifically, we examine its ability to recover the true gradient, its sensitivity to hyperparameters such as step size and downscaling factor, the diversity of generated samples, and its query efficiency. For our experiments, we first compute the "true gradients" by backpropagating through the ElasticFace model using the test set images from the FFHQ dataset. Next, we estimate the gradients using ZORO under various configurations of step size and downscaling factor, resulting in black-box gradients. Finally, we assess the quality of the estimated gradients by measuring the cosine similarity between the "ground truth" gradients and the "black-box" gradients.

Table 5: Cosine similarity b/w black-box and white-box gradients across step sizes $\delta$.

| $\delta$ | Similarity ($\uparrow$) |
|---|---|
| 0.01 | 0.006 |
| 0.1 | 0.140 |
| 1.0 | 0.675 |
| 3.0 | **0.778** |
| 5.0 | 0.773 |
| 10.0 | 0.721 |

**Sensitivity to step size.** We vary the noise parameters $\delta$ across the factors $\{0.01, 0.1, 1.0, 3.0, 5.0, 10.0\}$. We note that the noise factor must be tuned carefully to ensure maximum performance. In Table 5, we find the optimal parameters under our settings to be $\delta = 3.0$.

**Effect of downscaling x.** For a given downsampled image, we randomly sample a subset of $k$ indices and approximate the black box gradient (9). In Fig. 3 we report the cosine similarity with the white-box gradient on the y-axis with varying $k$ on the x-axis. We observe that for lower query budgets, lower-resolution images yield the best cosine similarity, whereas for higher query budgets, higher-resolution inputs lead to a better estimate. For a given query budget k, the highest cosine similarity is achieved with the smallest resolution that has at least $k$ dimensions. Naturally, the original image at $64 \times 64$ resolution provides the best estimate of the gradient but requires at least $5,000$ queries to obtain a reasonable approximation. Furthermore, in Table 6, we observe that the

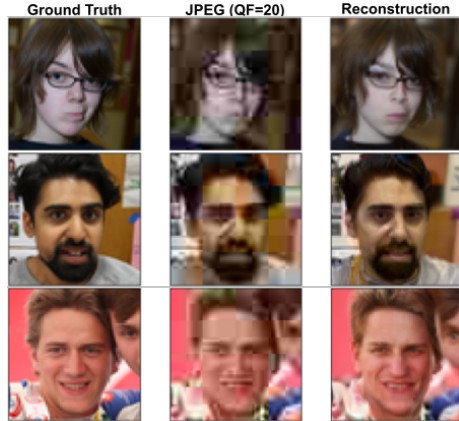

**Ground Truth**    **JPEG (QF=20)**    **Reconstruction**

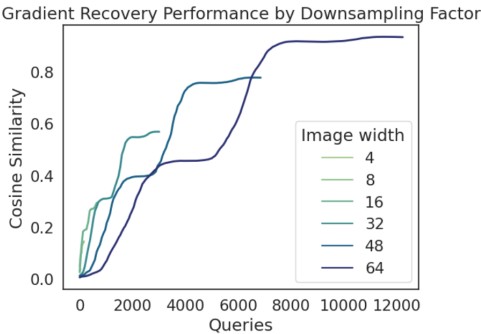

Figure 2: Results on JPEG restoration. From left to right: the ground truth image, the compressed JPEG image, restoration results from ZORO.

Figure 3: Gradient estimation performance when image is downsampled to a lower resolution. $64 \times 64$ resolution, when the image is not downsampled, provides the best gradient approximation at high query budgets but at low query budgets, yields the worst estimate of $\nabla_{\mathbf{x}}\ell(\mathbf{x})$.

Table 6: Ablations for different components of ZORO, specifically the guidance strengths: $\alpha_{\mathrm{f}}, \alpha_{\mathrm{c}}$, and the downscaling type on the test set of FFHQ. The error bars indicate standard deviation.

| ZORO Configuration | | | Metrics | | | |
|---|---|---|---|---|---|---|
| $\alpha_{\mathrm{f}}$ | $\alpha_{\mathrm{c}}$ | Downscaling | **Emb** ($\uparrow$) | PSNR ($\uparrow$) | SSIM ($\uparrow$) | LPIPS ($\downarrow$) |
| 0 | 0 | — | $0.52_{\pm 0.12}$ | $8.99_{\pm 1.21}$ | $0.14_{\pm 0.03}$ | $0.23_{\pm 0.00}$ |
| 10 | 1000 | Auto Encoder | $0.56_{\pm 0.09}$ | $10.94_{\pm 2.31}$ | $0.25_{\pm 0.03}$ | $0.23_{\pm 0.00}$ |
| 10 | 0 | Downsampling | $0.65_{\pm 0.03}$ | $10.42_{\pm 2.19}$ | $0.25_{\pm 0.03}$ | $0.181_{\pm 0.00}$ |
| 0 | 1000 | Downsampling | $0.72_{\pm 0.04}$ | $11.28_{\pm 1.75}$ | $0.35_{\pm 0.05}$ | $0.15_{\pm 0.00}$ |
| 10 | 1000 | Downsampling | $\mathbf{0.79}_{\pm 0.04}$ | $\mathbf{11.28}_{\pm 1.75}$ | $\mathbf{0.36}_{\pm 0.06}$ | $\mathbf{0.15}_{\pm 0.00}$ |

diffusion model produces significantly better inversions when using our proposed downsampling technique compared to using an autoencoder (Liu et al., 2020) for dimensionality reduction.

**The error correction strategy (Sec. 3.2) plays a significant role in the success of ZORO**. As shown in Suppl. 8, when the input is downsampled to a $16 \times 16$ resolution, both the autoencoder and bilinear-downsampling methods alone can barely recover the true gradients, achieving gradient cosine similarities approximately equal to zero. However, when bilinear-downsampling is combined with error correction, the cosine similarity improves significantly to $0.28$. Furthermore, for downsampling resolutions of $32 \times 32$ and $48 \times 48$, we observe that the error correction strategy greatly enhances the gradient similarity. Specifically, at the $32 \times 32$ resolution, the cosine similarity increases from $0.01$ (without error correction) to $\mathbf{0.59}$ (with error correction), and at the $48 \times 48$ resolution, it improves from $0.09$ to $\mathbf{80}$.

**Diversity of sampled images.** We qualitatively illustrate the diversity of sampled images during face embedding inversion, showcased in Fig. 1. We see that for different noise initializations, our model generates remarkably similar faces, with variations in certain features such as background and accessories. For example, consider the input image in the third row, right column, depicting an individual wearing a red shirt. Our model generates five different reconstructions of this individual, all wearing shirts of different colors.

## 5 RELATED WORK

**Diffusion models for inverse problems.** Diffusion models have been successfully applied to solve a variety of inverse problems, such as text-to-image generation (Rombach et al., 2021; 2022), inpainting (Chung et al., 2022b), colorization (Chung et al., 2022a), and super-resolution (Chung et al., 2023). Song et al. (2023) use classifier guidance for non-differentiable functions $f$, but unlike our method, they require defining a pseudo-inverse $f^{\dagger}$ for the non-invertible function $f$. However, such a

method is not applicable to tasks like embedding inversion, where the function $f$ is parameterized by a neural network and defining $f^{\dagger}$ is non-trivial.

**Zero-order methods.** Zero-order methods have been proposed for generating adversarial examples for vision classifiers (Chen et al., 2017; Andriushchenko et al., 2020). Tu et al. (2020) show that downsampling the input with an autoencoder can reduce the number of black-box queries required without compromising performance. In contrast to these works, we propose the use of zero-order methods for classifier-based guidance in diffusion models. More recently, zero-order methods have been proposed for other tasks, such as model stealing (Chen et al., 2019) and training large models (Chen et al., 2023). We are the first to apply zero-order methods to provide classifier-based guidance to diffusion models.

**Model inversion.** Inverting the representations of neural networks has been widely explored, especially in computer vision (Mahendran & Vedaldi, 2014; Dosovitskiy & Brox, 2016), as discussed in the general case by Morris et al. (2023). Zhmoginov & Sandler (2016) was the first to attempt inverting embeddings from facial attribute embedding models specifically. Most recently, diffusion models with classifier-free guidance have been proposed for solving this problem (Kansy et al., 2023).

## 6 DISCUSSION AND CONCLUSION

We introduced ZORO, a method that enables classifier guidance for diffusion models when only black-box access to the classifier is available. ZORO approximates the gradients through the black-box classifier using zero-order methods. We propose a novel error correction strategy in Sec. 3.2 that improves the scalability of the zero-order method without the need for additional neural networks, unlike prior works.

We demonstrate the application of ZORO in two key areas: first, in solving black-box inverse problems where the gradient $\nabla_{\mathbf{x}} \ell(f(\mathbf{x}), \mathbf{c})$ is defined but inaccessible (e.g., when the model is only accessible via an API), and second, in providing non-differentiable guidance in JPEG restoration, where ZORO outperforms all baselines and matches the performance of the *oracle*. Additionally, our method could potentially be useful for guiding diffusion models in scenarios where gradients are undefined—such as with discrete structures like molecules and text (Nisonoff et al., 2024)—as long as some signal is available to guide the model toward generating better solutions. We leave this exploration for future work.

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

## CONTENTS

# Appendices

## APPENDIX A    SAMPLER

Our sampler is described in Algorithm 1.

---

**Algorithm 1** Modification of the deterministic Heun sampler proposed in Karras et al. (2022).

1: **procedure** SAMPLER($D_\theta(\boldsymbol{x};t)$, $\mathbf{c}$, $q_\phi(\mathbf{c}|\mathbf{x})$, $t_{i\in\{0,\dots,T\}}$, $\alpha_{\mathrm{f}}$, $\alpha_{\mathrm{c}}$)
2:     **sample** $\boldsymbol{x}_T \sim \mathcal{N}\left(\mathbf{0},\ t_0^2\ \mathbf{I}\right)$
3:     **for** $i \in \{T-1,\dots,0\}$ **do**
4:         **sample** $\boldsymbol{\epsilon}_i \sim \mathcal{N}\left(\mathbf{0},\ S_{\mathrm{noise}}^2\ \mathbf{I}\right)$
5:         $u \leftarrow (1-\alpha_{\mathrm{f}})D_\theta(\boldsymbol{x}_i;\emptyset;t_i) + \alpha_{\mathrm{f}}D_\theta(\boldsymbol{x}_i;\mathbf{c};t_i) + \alpha_{\mathrm{c}}\hat{\nabla}_{\mathbf{x}}\log q_\phi(\mathbf{c}|\mathbf{x})$       ▷ Using (10).
6:         $\boldsymbol{d}_i \leftarrow \left(\boldsymbol{x}_i - u\right)/t_i$             ▷ Evaluate $\mathrm{d}\boldsymbol{x}/\mathrm{d}t$ at $t_i$
7:         $\boldsymbol{x}_{i-1} \leftarrow \boldsymbol{x}_i + (t_{i-1}-t_i)\boldsymbol{d}_i$         ▷ Take Euler step from $t_i$ to $t_{i-1}$
8:         **if** $t_{i-1} \neq 0$ **then**
9:             $\boldsymbol{d}_i' \leftarrow \left(\boldsymbol{x}_{i-1} - D_\theta(\boldsymbol{x}_{i-1};t_{i-1})\right)/t_{i-1}$       ▷ Apply 2$^{\mathrm{nd}}$ order correction
10:        $\boldsymbol{x}_{i-1} \leftarrow \boldsymbol{x}_i + (t_i - t_{i-1})\left(\frac{1}{2}\boldsymbol{d}_i + \frac{1}{2}\boldsymbol{d}_i'\right)$
11:     **return** $\boldsymbol{x}_T$

---

## APPENDIX B   EXPERIMENTAL DETAILS

We generally follow the training recipe from Karras et al. (2022) for training a diffusion model on the FFHQ dataset (Karras et al., 2019), which consists of $70,000$ images of human faces released under the Creative Commons license; as is standard in the literature, we split our data into $60,000$ train and $10,000$ test images. Note that we do not need paired facial data, only one face per individual, as we are not training a facial embedding model. We downsample images from their original resolution of $1024 \times 1024$ to $64 \times 64$ for training.

The model architecture is based on the UNet of Song et al. (2020). We use a batch size of 256, a learning rate of $2e-4$ with linear warm-up. We use the AdamW optimizer Loshchilov & Hutter (2019) and set $\beta_1 = 0.9, \beta_2 = 0.999, \epsilon = 1e-8$. We train for $200M$ images on 8 NVIDIA V100 GPUs, which takes about six days in total. We train a diffusion model to approximate $\mathbf{x}_0$ in the reverse process using a neural network, $D_\theta(\mathbf{x}_t, t)$, with parameters $\theta$. Following Karras et al. (2022); Song et al. (2020), the score function for the Variance Exploding type diffusion model in consideration is given by $\mathbf{s}_\theta(\mathbf{x}_t, t) = -(\mathbf{x}_t - D_\theta(\mathbf{x}_t, t))/\sigma_t^2$. Furthermore, the denoising network $D_\theta(\mathbf{x}, t) = c_{\mathrm{skip}}(t)\mathbf{x} + c_{\mathrm{out}}(t)F_\theta(c_{\mathrm{in}}(t)\mathbf{x}, c_{\mathrm{noise}}(t))$ where $F_\theta$ is the neural network to be trained, we use the values $c_{\mathrm{skip}}(t) = 1/(4t^2+1), c_{\mathrm{out}} = t/\sqrt{(4t^2+1)}, c_{\mathrm{in}}(t) = 2/\sqrt{(4t^2+1)}$. The scaling functions are represented by, $c_{\mathrm{skip}}(t), c_{\mathrm{out}}(t), c_{\mathrm{in}}(t), c_{\mathrm{noise}}(t) : \mathbb{R}^+ \to \mathbb{R}^+$. The loss function to train the diffusion model is given by $\mathcal{L}(\mathbf{x}_t, t) = \frac{4t^2+1}{t^2}\|D_\theta(\mathbf{x}_t, t) - \mathbf{x}_0\|_2^2$.

## APPENDIX C   ADDITIONAL EXPERIMENTS

### C.1   JPEG RESTORATION

By enabling differentiation through previously non-differentiable functions, we can now solve challenging inverse problems, provided we can efficiently sample from them. JPEG restoration serves as a practical example of such a problem. Images are often corrupted by excessive JPEG compression, resulting in degradation that is difficult to reverse. While one could train a purpose-built JPEG decompression algorithm, we propose to instead use a large, powerful, pretrained diffusion model to undo this corruption.

In fact, we can adapt ZORO to recover the original image by leveraging the natural image prior embedded within diffusion models. For these experiments, we subtly modify our sampling procedure. Instead of starting the sampling process from timestep $t = T = 100$, we begin by corrupting the low-quality JPEG image to timestep $t = 30$ and then run the sampling process with $\alpha_{\mathrm{c}} = 1$. Because $\mathbf{c}$ already contains significant information about the input, it is unnecessary to start the sampling process from pure Gaussian noise at $T = 100$.

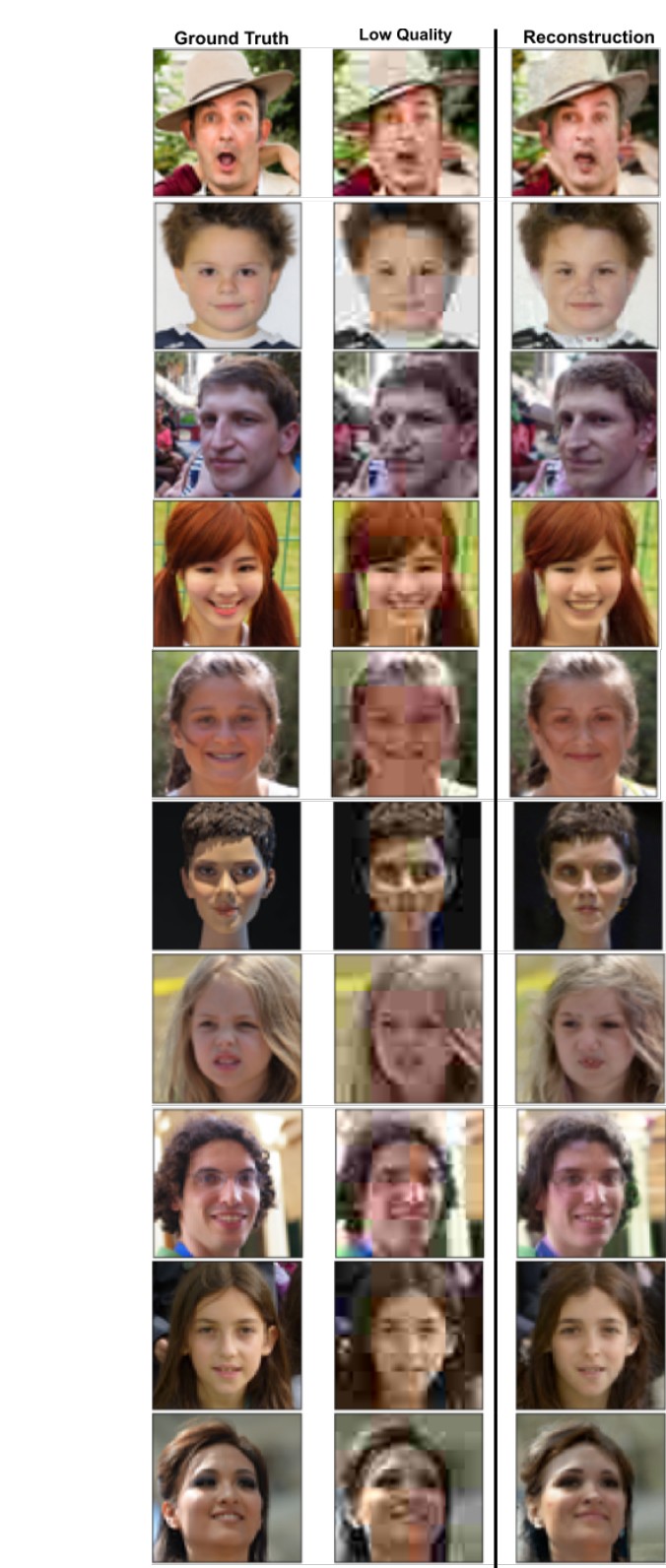

Figure 4: Randomly picked examples for JPEG Restoration experiments from the test set of FFHQ.

## C.2 DIRECTION-WISE ZERO-ORDER ESTIMATION

Another approach to approximating $\hat{\nabla}_{\mathbf{x}}\ell(\mathbf{x})$ may be to use a scaled random gradient estimator:

$$\hat{\nabla}_{\mathbf{x}}\ell(\mathbf{x}) = \sum_{i=1}^{n} \frac{\ell(\mathbf{x} + \beta\mathbf{u}) - \ell(\mathbf{x})}{\beta}\mathbf{u}, \tag{11}$$

where $n$ is the number of samples drawn, $\beta > 0$ is a smoothing parameter, and $\mathbf{u}$ is a vector drawn from a unit Euclidean sphere. Note that the value of this quantity tends to the true gradient as $\beta \to 0^+$.

This formulation allows us to smoothly trade queries to $\ell$ for better approximations of $\nabla_{\mathbf{x}}\ell(\mathbf{x})$. However, it is known to suffer from high variance and slow convergence (Liu et al., 2020). We considered these two estimators in our experiments in Sec. 4, and analyze their performance in-depth in Sec. 4.4.

**Sensitivity to step size.** We vary the noise parameters ($\beta$ for Directional and $\delta$ for Coordinate) across the factors $\{0.01, 0.1, 1.0, 3.0, 5.0, 10.0\}$. We note that this factor has a large impact on the results across the board: regardless of the choice of gradient estimation algorithm, and the noise factor must be tuned carefully to ensure maximum performance. In Table 7, we find the optimal parameters under our settings to be $\beta = 0.1$ for Direction-Wise estimation and $\delta = 3.0$ for Coordinate-Wise.

Table 7: Gradient estimator comparison across step sizes. We grid search across step sizes ($\delta$ for coordinate-wise estimation or $\beta$ for direction-wise estimation). **Higher** is better.

| Step size | Coordinate | Directional |
|-----------|-----------|-------------|
| 0.01 | 0.006 | 0.224 |
| 0.1 | 0.140 | **0.580** |
| 1.0 | 0.675 | 0.542 |
| 3.0 | **0.778** | 0.375 |
| 5.0 | 0.773 | 0.246 |
| 10.0 | 0.721 | 0.091 |

**Gradient Estimation Quality** Fig. 5 depicts the query efficiency of both methods. We observe that our proposed coordinate-wise gradient estimator is more efficient in approximating the true gradients for a given query budget.

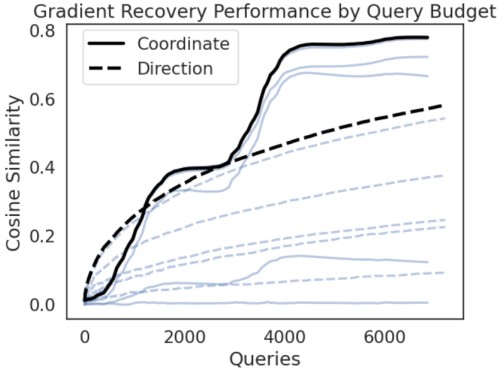

Figure 5: Query efficiency of various gradient estimators measured on test-set images embedded with ElasticFace, using cosine similarity loss. Black lines represent the best hyperparameter configurations for each method. Coordinate-Wise gradient estimation outperforms Direction-Wise starting around 1000 queries.

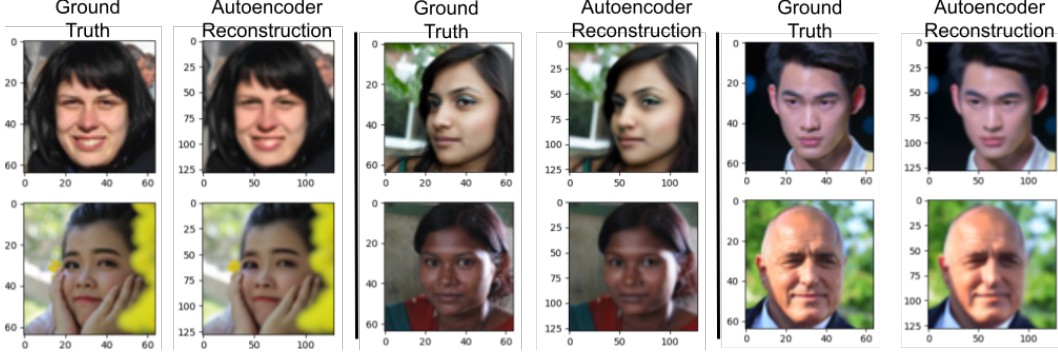

Figure 6: Input reconstructions by the autoencoder. The pre-trained model almost accurately reconstructs the original image. The original input has a $64 \times 64$ resolution. It is upsampled to $128 \times 128$ to match the input specification resolution for the autoencoder. We observe that the autoencoder almost perfectly reconstructs the input.

### C.3 AUTOENCODER INPUT DOWNSCALING

Instead of training an autoencoder from scratch we use a pre-trained model provided by Stability AI's SDXL-VAE[1], a latent diffusion model where the diffusion occurs in the pretrained, learned latent space of an autoencoder. This specific VAE model was chosen as it outperforms similar models on various metrics such as rFID, PSNR, SSIM, and PSIM. Furthermore, as a sanity check, we ensure that the autoencoder almost perfectly recovers the input in Fig. 6.

### C.4 SYNTHETIC EXPERIMENTS

We provide visualizations of the ground truth image, the corrupted image, and their reconstructions for corruption types: (1) Gaussian Blur & Grayscale in Fig. 8, (2) Gaussian Blur & Mask Fig. 9, and (3) Grayscale & Mask in Fig. 7 respectively.

### C.5 ZORO ABLATIONS

FFHQ samples generated by ZORO with only CFG Fig. 10 and w/o any guidance Fig. 11.

---

[1]https://huggingface.co/stabilityai/sdxl-vae

972
973
974
975
976
977
978
979
980
981
982
983
984
985
986
987
988
989
990
991
992
993
994
995
996
997
998

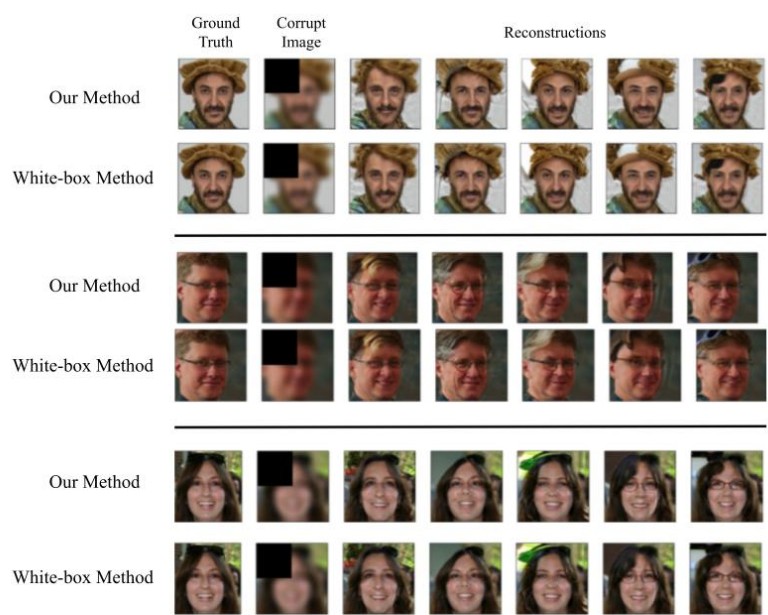

Figure 7: Applying Gaussian Blurring and Masking corruptions.

999
1000
1001
1002
1003
1004
1005
1006
1007
1008
1009
1010
1011
1012
1013
1014
1015
1016
1017
1018
1019
1020
1021
1022
1023
1024
1025

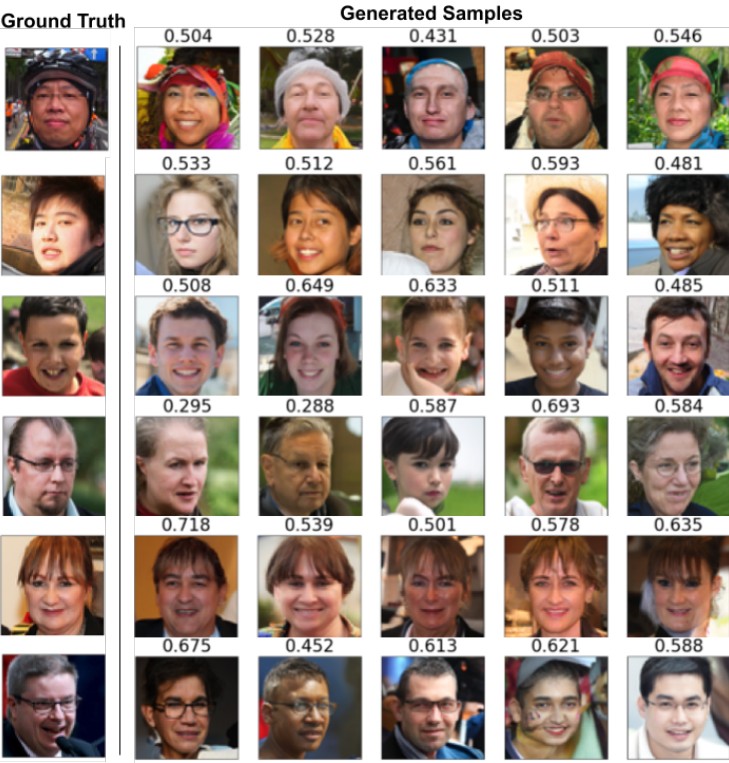

Figure 10: Samples generated by ZORO with only Classifier Based Guidance $\alpha_{\mathrm{f}} = 10, \alpha_{\mathrm{c}} = 0$. The numbers at the top of each sample denote the cosine similarity between the embeddings of the ground truth image and the sample.

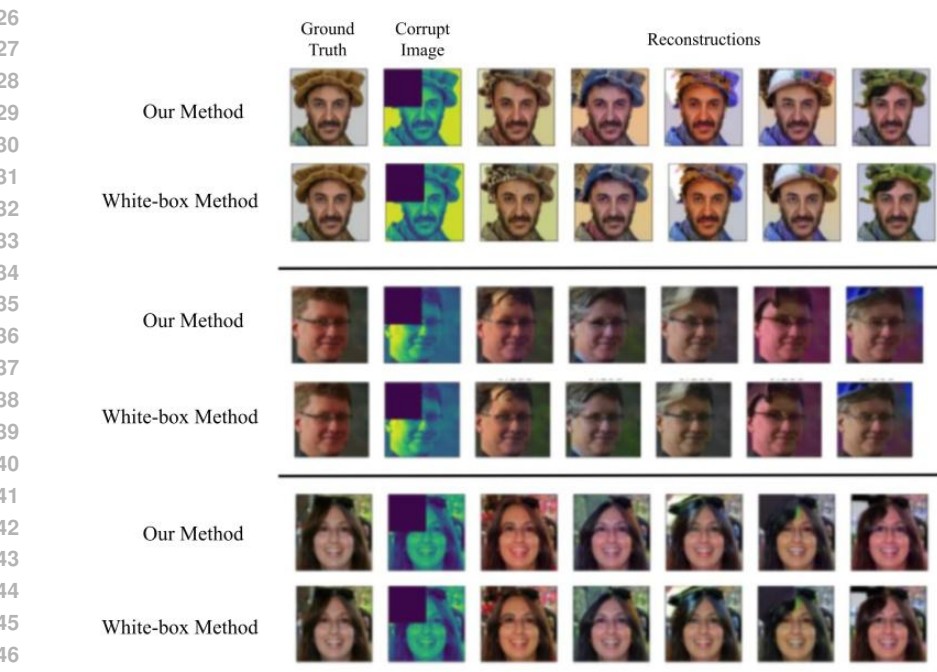

Figure 8: Applying Grayscale and Masking corruptions.

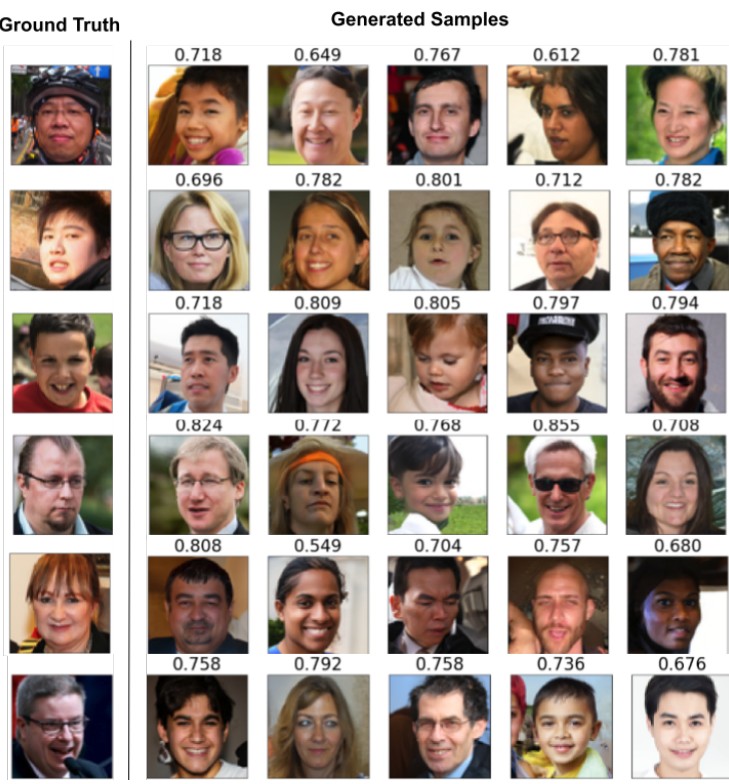

Figure 11: Samples generated by ZORO w/o guidance, i.e., $\alpha_f = 0, \alpha_c = 0$. The numbers at the top of each sample denote the cosine similarity between the embeddings of the ground truth image and the sample.

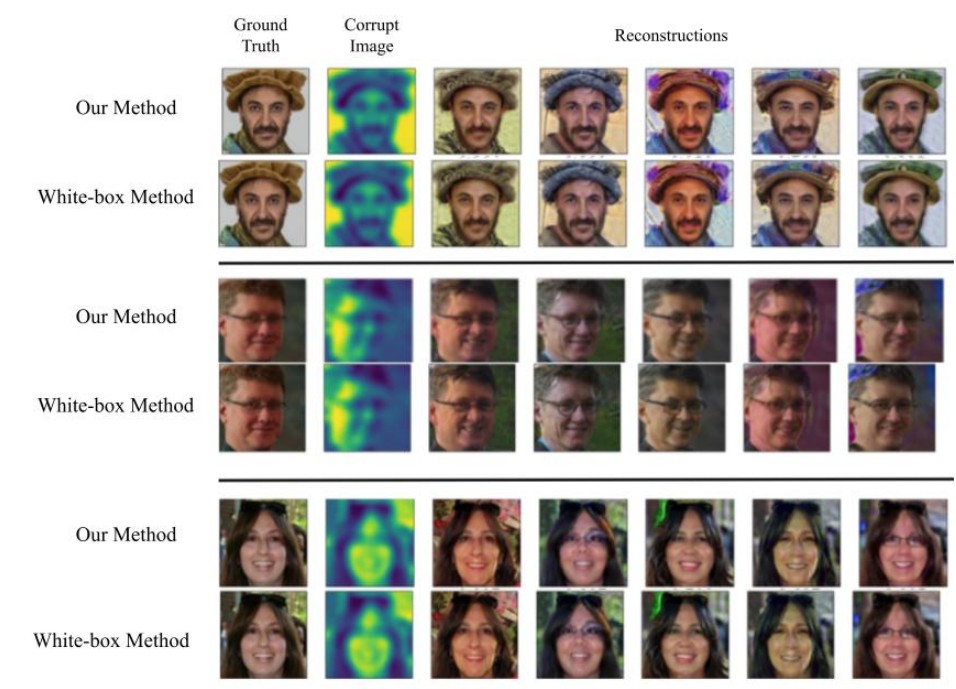

Figure 9: Applying Gaussian Blurring and Grayscale corruptions.

## C.6 ERROR CORRECTION

Table 8 highlights the importance of error correction (9) in scaling up ZORO.

Table 8: Cosine similarities between white-box and black-box gradients on a batch of 128 FFHQ images on the test set. **Higher** is better. The setup for the autoencoder is described in Suppl. C.3.

| Downscaling Type | $16 \times 16$ | $32 \times 32$ | $48 \times 48$ |
|---|---|---|---|
| AE | 0.01 | — | — |
| Downsampling | 0.01 | 0.01 | 0.09 |
| Downsampling + Error Correction (9) (**Ours**) | **0.28** | **0.59** | **0.80** |

