# OpenReview forum: "Zero-Order Diffusion Guidance for Inverse Problems"
_ICLR.cc/2025/Conference — ICLR 2025 Conference Withdrawn Submission_

### Official Review · Reviewer_tbQh · 2024-10-28

**Soundness:** 1
**Presentation:** 1
**Contribution:** 2
**Rating:** 3
**Confidence:** 4

**Summary:**

The authors propose ZORO, a method for solving inverse imaging problems using diffusion models in a setting where only forward passes can be applied to the pre-trained model.

**Strengths:**

- The combination of classifier-free and classifier-based guidance.

- The scaling up of the gradient estimation and its Motivating examples in Section 3.2.

**Weaknesses:**

- The paper is not well-written and hard to follow with several notational issues. See Major and Minor comments/questions below for examples.

- The problem is not very well-motivated. Why don't we have access to the pre-trained model? It is a generative model, employed as a prior. For example, before diffusion models, can the authors find a practical application in inverse problems where the prior is not fully accessible? I have not seen such scenario or paper, but if the authors can find any, it would help motivate the paper. I know authors say security applications, but this should be better explained with more practical applications/discussions (and citations).

**Questions:**

## Major Comments/Questions:

- The forward model in 131 is typically defined as $c = f(x_0)+noise$. Is noise considered here?

- Where is the discrete case? I feel that the most practical case for the setting of the paper is when exact gradients can not be computed, which has many applications as the authors mention. However, I don't see the proposed method tried in such scenarios.

- The use of the word "adversary" in the abstract is strange. For black box attacks on Neural Networks, the use of "adversary" makes sense because the goal of the attacker is to generate adversarial examples that fool the classifier without accessing the gradients (or only can perform forward passes to the network). But in this paper, the assumption is that we can only make forward passes to the DM network. The purpose is completely different.

- Citations, support, and/or discussion are needed here "Out of the box, zero-order methods are slow". This is lines 149 to 150.

- $l$ was initially defined to take $c$ and $f(x_0)$ which means that it is $\mathbb{R}^m \times \mathbb{R}^m \rightarrow \mathbb{R}_+$, then a different mapping is defined later in line 154. The authors afterwards used $l$ with different arguments.

- In line 103, the score function is typically defined as the mapping $\mathbb{R}^n \times \{0,...,T\} \rightarrow \mathbb{R}^n$. It takes $t$ in the second argument of the function like in Equation (3).

- Estimating the gradients (or finite difference methods) have a long history and rich literature. What do authors mean when they say "novel"? Do the authors mean they adopt an existing method for estimating gradients?

- Same above point applies to adopting bi-linear down- and up-sampling for dimensionality reduction. The authors can say that  we adopted these strategies in this particular setting.

- The authors do conduct ablation and sensitivity studies for ZORO. While the selection of the parameters to ablate for $\delta$ is convincing, the selection of $(\alpha_f,\alpha_c)$ is not convincing as the authors only consider $\alpha_f \in \{0,10\}$ and $\alpha_c \in \{0, 1000\}$.

- DMs are trained on downscaled images. Why is that? If it is because ZORO can not handle larger spaces, then this should be clarified and discussed earlier in the paper. This is a major limitation in ZORO. I am not against the fact that ZORO can only apply to lower dimensions (if it is the case), but i believe this should be stated and discussed clearly.

- While the authors report qualitative results against the oracle and other baselines, run-time and NFEs are not reported in the tables.

- Insufficient baselines: Even with the reduction of the FFHQ dataset and the trained DM in the paper, why the oracle isn't Diffusion Posterior Sampling (as an example of a leading DM-based Inverse Problem Solver)? DPS only requires a simple gradient step at each sampling iteration. I am not saying that ZORO must outperform (or even be on par) all DM-based methods (see 'https://arxiv.org/abs/2410.00083'), but it should be compared to existing leading methods.

- In section 4.1, it was never mentioned how the synthetic dataset was generated. Appendix B does not have it either. Furthermore, the write-up for Section 4.1 never refers to Table 1 which include the results for this experiments.

- Does the "query count" account for diffusion sampling?

## Minor Comments/Questions:

- Repeated sentences "we may be working with sensitive information from data obfuscated by a black-box program" in lines 36 (Introduction) and 144 (section 3). Another example is "Our key idea is to approximate classifier-based guidance by using zero-order gradient estimators".

- Is $\mathcal{D}$ the training set or a distribution? $p_0$ is later used for distribution.

- Why cosine similarity is used in Figure 1? Typically, SSIM, PNSR, and LPIPS are used. To compare the distributions of reconstructed and ground truth images, FID might be used (if a sufficient amount of test data points is accessible). Furthermore, can the authors include the degraded images?

- In Figure 1, what is the difference between row 1 and columns 2 to 6? For the introduction section, this is not explained. A brief description is required here.

- A dot after Equation (1).

- No $i$ in the numerator of the equation in line 158. It should be $u_i$.

- Can't the authors use limit to describe the nablas in line 161?

- Isn't $f(x)$ the forward operator? what is $\partial f / \partial x$ in line 163?

- What is $h$ in line 170? There is no $h$ is Equation (6). $h$ was only defined in subsequent discussion.

- It is recommended that the JPEG restoration visualizations (for example Figure 4) to include baselines and LPIPS (and PSNR) scores.

- Equation (9) or Equation 9 in line 240.

- In line 299, what do you mean by "we train by 200M images"?

- In lines 310 to 312, the authors say "We observe that ZORO nearly matches the performance of oracle qualitatively and quantitatively **as shown** in Sec. 1.". Do you mean as stated in Sec.1? Where in Section 1 the authors show this?

- $\alpha_u$ is missing in Algorithm 1.

---

### Official Review · Reviewer_jzGF · 2024-10-28

**Soundness:** 2
**Presentation:** 2
**Contribution:** 2
**Rating:** 5
**Confidence:** 4

**Summary:**

This paper proposes an approach to solve inverse problems without direct knowledge of the forward model. Using only access to the model outputs authors leverage zero-order gradient estimation combined with an up/down-sampling technique to find a solution to inverse problems, including JPEG restoration and model inversion for face reconstruction.

**Strengths:**

- The paper is well motivated. There has been a large body of work on solving inverse problems with access to a differentiable forward model, however black box access problems have been much less studied.
- The proposed technique is interesting and provides a sensible way to reduce the complexity of zero-order optimization.
- The paper is fairly well written and can be easily followed for most parts.

**Weaknesses:**

- The experiments are on low-resolution (64x64) images. It is unclear how feasible the proposed method is in higher dimensions, including standard image resolutions.
- The proposed dimensionality reduction technique is not properly compared to other sensible baslines (see under Questions).
- There are some unclear parts in the experiments, further detailed under Questions.

**Questions:**

- How does the performance of ZERO scale to standard image dimensions (256x256 and above)?
- It is somewhat unclear why the autoencoder underperforms the proposed down/upsampler in dimensionality reduction. The autoencoder should provide better compression as it has been trained on the target dataset. How does the autoencoder perform if we add the correction term, similarly to the proposed technique?
- How have the hyperparameters (different guidance weights) been tuned? Is $\alpha_f$ set to $0$ or $500/1000$ (possible typo)?
- What is the compute cost of inversion? How much time does a sample take?
- Why is there no CFG in the JPEG restoration experiments?
- In Table 5, the sensitivity of gradient estimation to the step size is depicted. However, what would be more interesting to see is how it impacts performance.
- Why are there 'steps' in the cosine similarity plots in Figure 3?
- The effect of the error correction term I believe is important and the corresponding experiment could be moved from the appendix to the main paper.

---

### Official Review · Reviewer_ujjw · 2024-11-01

**Soundness:** 2
**Presentation:** 2
**Contribution:** 1
**Rating:** 3
**Confidence:** 3

**Summary:**

This paper introduces a derivative-free, diffusion-based, plug-and-play approach for tackling inverse problems. Specifically, the authors decompose the score of the posterior distribution into an unconditional score and the conditional probability score $\nabla_x \log p_t (x|c)$. To address the cases where the conditional probability's derivative is unavailable, the paper estimates it through a finite difference method (FDM) applied individually to each coordinate of $x$. Given the computational expense of this directional derivative in high-dimensional spaces, the authors incorporate a downsampling and upsampling module, $f$ and $g$, to apply FDM in a reduced space. They further refine certain approximations to minimize estimation error. The proposed model is evaluated on facial embedding inversion and JPEG restoration tasks.

**Strengths:**

The paper is well-structured, making the proposed approach straightforward to understand. Additionally, the evaluation metrics provided are comprehensive.

**Weaknesses:**

- The primary contribution of this work is the use of finite difference methods (FDM) to approximate the gradient of the non-differentiable degradation operator $f$. However, this contribution may be somewhat limited. For instance, FDM-based gradient approximation is used as a baseline in related work [1] (DPS-fGSG, DPS-cGSG). While I recognize that direct comparison with [1] is not necessary since it is a concurrent work, it is worth noting that approximating non-differentiable operators with FDM is not an entirely novel approach.

- Approximating directional derivatives in this way remains computationally costly, which could limit the scalability of the method.

- Furthermore, I am concerned that this approach may not perform effectively when $f$ is highly sensitive to small changes, as could occur in full-waveform inversion or PDE-based inverse problems, as discussed in [1].

- Significant improvements are also needed in the experimental section. First, there are no comparisons with standard inverse problems, such as deblurring, inpainting, and super-resolution. Although the paper focuses on non-differentiable inverse problems, comparisons with standard applications are essential to assess the method’s general effectiveness, even if its performance is lower in those cases.

- Second, several existing methods address non-differentiable operators, such as DPG [2], SCG [3], FPS [4], and DPS-GSG, yet the paper does not provide comparative evaluations with these approaches.

- Third, the paper lacks examples where $f$ is a true black-box or non-differentiable operator. I strongly recommend that the authors include applications with non-differentiable operators, as seen in [1, 2, 3, 5], and perform comparisons with some of these methods.

**Questions:**

- Since the proposed method estimates the gradient of $f$ using the finite difference method, one would expect performance to improve as the discretization step $\delta$ decreases. Has this relationship been explicitly discussed in the paper? Additionally, Table 5 suggests that the method may not perform well when $\delta$ is small.

- Table 6 indicates that the algorithm is highly sensitive to the guidance rate, suggesting a dependency on numerous hyperparameters that require fine-tuning.

- Is there any theoretical justification provided for the transition from Equation 7 to Equation 9?

- In Equation 5, replacing “=” with $\approx$ might better reflect the intended approximation.

$ $

References

[1] InverseBench: Benchmarking Plug-and-Play Diffusion Models for Scientific Inverse Problems, preprint, 2024.

[2]  Tang et al.,, Solving General Noisy Inverse Problem via Posterior Sampling: A Policy Gradient Viewpoint, AISTATS, 2024.

[3] Huang et al., Symbolic Music Generation with Non-Differentiable Rule Guided Diffusion, ICML, 2024.

[4] Dou et al., Diffusion Posterior Sampling for Linear Inverse Problem Solving: A Filtering Perspective, ICLR, 2024.

[5] Zheng et al., Ensemble Kalman Diffusion Guidance: A Derivative-free Method for Inverse Problems, preprint, 2024.

---

### Official Review · Reviewer_hT9w · 2024-11-03

**Soundness:** 2
**Presentation:** 2
**Contribution:** 1
**Rating:** 3
**Confidence:** 5

**Summary:**

This paper proposes a supervised method to solve inverse problems without the access to the gradients of the degradation function. To achieve this, this paper employs a coordinate-wise gradient estimation technique based on finite differences to approximate the true gradient, and then uses an autoencoder-like architecture to project images to a lower-dimension space to scale up the gradient estimation process.

**Strengths:**

1. The structure of this paper is clear to readers.

**Weaknesses:**

1. Lack of important literature review and questionable settings. This paper tries to solve inverse problems when the gradients of forward functions are unknow, which seems to be interesting, but there are many existing inverse problem solvers that even assume the forward functions are not unavailable. Dozens of the supervised methods reviewed in [1] only assume the pair (X_i, Y_i) are given. In this sense, such existing methods should be reviewed and compared. More importantly, given the methods that even assumes that the forward functions are unknown have already made significant successes under the supervised settings, I do not think that the setting discussed in this paper has enough values. Personally, I strongly encourage the authors to conduct a comprehensive literature review in future research endeavors and switch this paper to a zero-shot setting since most of zero-shot diffusion-model-based methods assume both the forward functions and the gradients of forward functions are known, as discussed in [1].
2. Lack of contribution. This paper claims to approximate the truth gradient using the coordinate-wise gradient estimation technique based on finite differences in Section 3.1. However, this is a very standard technique that has already been studies in [2,3,4].
3. Lack of novelty. In Section 3.2, this paper proposes to use an autoencoder structure to project data into lower-dimensional space to scale up the coordinate-wise gradient estimation, which is a very common technique.
4. Lack of reasoning. In Section 3.2.1, this paper uses bilinear downsampling and upsampling to reduce the dimension of the data, but there is no clear evidence discussed why to choose this heuristics intead of other approaches; in Section 3.2.2, this paper does not provide enough insight and derivation to obtain this error correction term although limited ablation studies are conducted in Table 8.
5. Lack of empirical evidence. This paper proposes a method for solving inverse problems, but it only considers two uncommon inverse problems, i.e., reconstructing faces from embeddings and JPEG restoration. If this paper wants to claim a general method for dealing with inverse problems, then more applications should be considered, e.g., denoising, super-resolution, inpainting, deblurring and colorization. In addition, as I mentioned in the Weakness 1, it is essential to compare the proposed method with various supervised techniques, especially since these methods do not even require prior knowledge of the forward functions. And I expect that the proposed method should beat these methods since this paper assumes the access to the forward functions although the gradients are unavailable.






[1] Li, X., Ren, Y., Jin, X., Lan, C., Wang, X., Zeng, W., Wang, X. and Chen, Z., 2023. Diffusion Models for Image Restoration and Enhancement--A Comprehensive Survey. arXiv preprint arXiv:2308.09388.

[2] Lian, X., Zhang, H., Hsieh, C.J., Huang, Y. and Liu, J., 2016. A comprehensive linear speedup analysis for asynchronous stochastic parallel optimization from zeroth-order to first-order. Advances in Neural Information Processing Systems, 29.

[3] Berahas, A.S., Cao, L., Choromanski, K. and Scheinberg, K., 2022. A theoretical and empirical comparison of gradient approximations in derivative-free optimization. Foundations of Computational Mathematics, 22(2), pp.507-560.

[4] Chen, A., Zhang, Y., Jia, J., Diffenderfer, J., Liu, J., Parasyris, K., Zhang, Y., Zhang, Z., Kailkhura, B. and Liu, S., 2023. Deepzero: Scaling up zeroth-order optimization for deep model training. arXiv preprint arXiv:2310.02025.

**Questions:**

See weaknesses.

**Details Of Ethics Concerns:**

None.

---

### Official Review · Reviewer_kV2v · 2024-11-04

**Soundness:** 3
**Presentation:** 3
**Contribution:** 2
**Rating:** 6
**Confidence:** 4

**Summary:**

The authors propose a zero-order optimization method for classifier-conditional diffusion models when used as priors for inverse problems. Specifically, this work focuses on inverse problems that do not have access to gradients, such as scenarios involving black-box access to face embeddings or data streamed from an API. The proposed method introduces a coordinate-based approximation of the gradient (as described in Equation 10) by estimating the gradient in a lower-dimensional space, which outperforms deep auto-encoders for this task. Furthermore, the authors empirically validate their method across a wide range of inverse problems, demonstrating impressive performance relative to other successful baselines.

**Strengths:**

1. **Motivation**

   The proposed method is well motivated by important applications that arise in real-world scenarios.

2. **Novel Approach**

   The authors introduce a novel approach for approximating the low-dimensional gradient using a downsampling function and an error correction strategy. This method leads to improved performance and reduced computational overhead compared to previous approaches that rely on learned autoencoders.

3. **Empirical Results**

   **a. Tables 1 and 2** – The proposed method performs similarly to the oracle and significantly outperforms the baselines.

   **b. Tables 3 and 4 and Figure 2** – For JPEG restoration, the proposed method demonstrates a significant improvement in reconstruction performance, both empirically and visually.

**Weaknesses:**

1. **Clarity in Section 3**:

   The writing in Section 3 can be somewhat confusing due to the inconsistent assignment of the variable \( c \). Specifically:

   - **Line 112**: \( c \) is defined as conditional information that guides the generation process.
   - **Line 117**: \( q_\phi(c|x) \) is described as the probability of attribute \( c \) given input \( x \), implying that \( c \) represents a class.
   - **Later in the Section**: \( c \) is defined as a measurement \( c \in \mathbb{R}^m \).

   This multiple redefinition of \( c \) can lead to confusion for readers trying to follow the methodology.

2. **Lack of Novelty in Some of the Proposed Methodology**:

   Some aspects of the proposed methodology lack originality. Coordinate-based gradient approximations are already commonly used for zero-order optimization. Additionally, the authors only make slight modifications to the sampler presented in Algorithm 1, which does not constitute a significant advancement over existing methods.

3. **Absence of Code and Implementation Details**:

   Although the paper is experimental in nature, the authors do not mention or provide any code. Providing code or implementation details is crucial for reproducibility and for other researchers to validate and build upon the work presented.

**Questions:**

1. **Clarification of Novelty of the Proposed Method*

   Although coordinate-based approximations of gradients are a well-established method, the authors should highlight the novelty of their approach and its contribution to the existing literature. Specifically, it would be beneficial to elaborate on how their method differentiates from or improves upon traditional coordinate-based gradient approximation techniques.

2. **Reproducibility Statement**

   The authors should provide a reproducibility statement to enhance the credibility and utility of their work. This statement could include details about the availability of code, datasets, and any specific implementation guidelines that would allow other researchers to replicate the results presented in the paper.

3. **Clarification on Variable \( c \) Reassignment**

   There is confusion regarding the reassignment of the variable \( c \) throughout Section 3. Please refer to the weakness section above for context.

---

### Note · Authors · 2024-11-25

I have read and agree with the venue's withdrawal policy on behalf of myself and my co-authors.